# 3D-Cultured MC3T3-E1-Derived Exosomes Promote Endothelial Cell Biological Function under the Effect of LIPUS

**DOI:** 10.3390/biom14091154

**Published:** 2024-09-13

**Authors:** Xiaohan Liu, Rui Cheng, Hongjuan Cao, Lin Wu

**Affiliations:** 1Department of Prosthodontics, School and Hospital of Stomatology, China Medical University, Shenyang 110002, China; 20162209@cmu.edu.cn (X.L.); rui.cheng@kcl.ac.uk (R.C.); 2Liaoning Provincial Key Laboratory of Oral Diseases, China Medical University, Shenyang 110002, China

**Keywords:** low-intensity pulsed ultrasound (LIPUS), bone regeneration, angiogenesis–osteogenesis coupling, exosomes

## Abstract

Porous Ti-6Al-4V scaffold materials can be used to heal massive bone defects because they can provide space for vascularisation and bone formation. During new bone tissue development, rapid vascular ingrowth into scaffold materials is very important. Osteoblast-derived exosomes are capable of facilitating angiogenesis–osteogenesis coupling. Low-intensity pulsed ultrasound (LIPUS) is a physical therapy modality widely utilised in the field of bone regeneration and has been proven to enhance the production and functionality of exosomes on two-dimensional surfaces. The impact of LIPUS on exosomes derived from osteoblasts cultured in three dimensions remains to be elucidated. In this study, exosomes produced by osteoblasts on porous Ti-6Al-4V scaffold materials under LIPUS and non-ultrasound stimulated conditions were co-cultured with endothelial cells. The findings indicated that the exosomes were consistently and stably taken up by the endothelial cells. Compared to the non-ultrasound group, the LIPUS group facilitated endothelial cell proliferation and angiogenesis. After 24 h of co-culture, the migration ability of endothelial cells in the LIPUS group was 17.30% higher relative to the non-ultrasound group. LIPUS may represent a potentially viable strategy to promote the efficacy of osteoblast-derived exosomes to enhance the angiogenesis of porous Ti-6Al-4V scaffold materials.

## 1. Introduction

There is an increasing demand for artificial bone materials to heal massive bone defects in patients after revision surgery, malignancy resection, infection, or severe trauma [1]. In recent years, there has been increased interest in porous scaffold materials. Scaffolds are a promising technique, acting as supportive structures while providing essential nutrients to facilitate cell attachment and growth on their surfaces [2]. They also create space for vascularisation and bone formation [3]. The long-term success of implanted ex vivo-generated tissue constructs depends significantly on rapid and sufficient integration with the host vasculature to ensure an adequate supply of oxygen and nutrients to the implant. However, the difficulty of inducing rapid vascular ingrowth during new bone tissue development is a major limitation of bone tissue engineering approaches for bone defect repair [4].

Low-intensity pulsed ultrasound (LIPUS), as a specialised ultrasound modality, propagates with low intensity and is emitted in pulsed form, enabling the targeted delivery of acoustic energy to tissues while minimising thermal effects during treatment [1]. LIPUS has been established as a non-invasive physical therapy modality with positive biological significance across all stages of fracture healing. In addition, it has been confirmed that LIPUS can promote osteoblast migration, proliferation, and differentiation and enhance bone ingrowth and bone formation in scaffolds [5,6]. However, traditional biomedical scaffolds are limited by the lack of vascularisation in synthetic bone structures, impacting implant survival and integration [7]. Numerous investigations have highlighted the significance of direct cell-to-cell communication among osteoblasts, endothelial cells, and their precursors, along with the autocrine and paracrine factors they secrete, in fostering rapid vascularisation and effective bone formation [4,8,9]. A prior study gathered conditioned media from LIPUS-treated human umbilical vein endothelial cells (HUVECs) co-cultured with osteoblasts, revealing an upregulation of osteogenic-related gene expression in osteoblasts compared to the control group. This finding confirms the influence of LIPUS on paracrine signalling and cell-to-cell communication in the context of angiogenesis–bone coupling. However, existing in vitro studies predominantly focus on osteoblasts or endothelial cells in isolation. As a result, the potential of osteoblasts within scaffold materials under LIPUS loading to influence vascularisation still requires further investigation.

Exosomes are extracellular bilayer lipid membrane vesicles between 30 and 150 nm in diameter, generated by endocytosis and budding as intraluminal vesicles in the luminal space of multivesicular bodies and secreted following the binding of multivesicular bodies with cell membranes [10]. Cells can release exosomes in nearly all physiological and pathological states, carrying proteins, mRNA, non-coding RNA, DNA, lipids, and other nutrient-dense substances [11]. Exosomes have the potential for multidirectional use. Exosomes can transmit differently expressed miRNAs between cells to impact intercellular communication and play an essential role in the paracrine coupling process of angiogenesis and osteogenesis, according to recent research [12,13]. Exosomes produced from a serum-free conditioned medium of human bone marrow mesenchymal stem cells stimulated early bone repair and angiogenesis in a rat model of femoral non-union while promoting cell migration and osteogenic and angiogenic gene expression in MSC cultured in vitro [14]. Exosomes generated from human umbilical cord mesenchymal stem cells and released from a glass/hydrogel scaffold stimulated vascularised bone regeneration by transferring miR-23a-3p, as reported by Hongxing Hu et al. [15]. Some research also indicates that LIPUS stimulation can enhance the release of exosomes from cells, potentially increasing cell-to-cell communication and the transfer of signalling molecules. Additionally, LIPUS may impact the cargo of exosomes, altering proteins, RNA, and other molecules they carry. By influencing the content of exosomes, LIPUS could potentially modulate the signalling pathways and biological functions mediated by these extracellular vesicles [5,6,16]. Therefore, LIPUS may enhance osteogenesis–vascular coupling by modulating the exosomes secreted by osteoblasts on Ti-6Al-4V scaffold materials. This indicates that exosomes derived from 3D-cultured MC3T3-E1 cells under LIPUS stimulation could represent a promising therapeutic strategy for treating bone defects.

## 2. Materials and Methods

### 2.1. Scaffold Preparation

A cylindrical three-dimensional porous Ti-6Al-4V scaffold was supplied by the Institute of Metal Research, Chinese Academy of Sciences. The material exhibited compressive properties [17]. A porous 3D Ti-6Al-4V scaffold with a diameter of 15 mm, height of 2 mm, pore size of 400–500 micrometres, and porosity of 65–70% was manufactured using an electron beam melting system (Arcam A1, Arcam, Mölndal, Sweden) as described in our previous work [18].

Before experimentation, the materials were cleaned in an ultrasonic cleaning machine, dried with filter paper, and sterilised under high-temperature and -pressure conditions (0.21 MPa, 134 °C). Before use, the materials were wetted in the culture medium using a vacuum device.

### 2.2. Cell Culture

Mouse pre-osteoblast precursor cells MC3T3-E1 were provided by the Central Laboratory of China Medical University. MC3T3-E1 cells were cultured in alpha-minimum essential medium (α-MEM, HyClone Laboratories Inc., Logan, UT, USA) containing 10% foetal bovine serum (FBS, Clark, Richmond, VA, USA) and 1% penicillin–streptomycin dual antibiotics (HyClone Laboratories Inc., Logan, UT, USA). The medium was changed every two days, and the cells were passaged when 80% confluence was reached. The cells used in the experiments were in passages 5 to 15.

Human umbilical vein endothelial cells (HUVEC) were provided by the Central Laboratory of China Medical University. HUVECs were cultured in DMEM/F12 complete medium (HyClone Laboratories Inc., Logan, UT, USA) containing 10% FBS and 1% penicillin–streptomycin dual antibiotics. The cells were cultured in a CO_2_ incubator at 37 °C with a cell density of 80% for passaging. The cells used in the experiments were in passages 3 to 5.

### 2.3. Cell Seeding

Pre-treated porous Ti6Al4V materials were placed in a 6-well plate, with 3 materials placed in each well. A 20 µL suspension of MC3T3-E1 cells at a density of 1 × 10^6^ cells/mL was seeded onto the materials and incubated in a cell culture incubator for 1 h to allow for cell adhesion. Subsequently, 3 mL of exosome-free complete culture medium was added to each well, and the plate was placed in a 37 °C, 5% CO_2_ cell culture incubator. After 24 h of culture, the samples were randomly assigned to the LIPUS group (loaded with ultrasound) and the control group (treated with sham irradiation under the same conditions).

### 2.4. LIPUS Ultrasonic Stimulation

LIPUS stimulation was applied using a Sonicator^®^ 740 (Mettler Electronics Corp, Anaheim, CA, USA) with a 5 cm^2^ ultrasonic transducer (ME 7413). Based on preliminary experiments by our research group, the ultrasound parameters were set as follows: frequency at 1 MHz, intensity at 30 mW/cm^2^, pulse width at 1 ms, pulse repetition frequency at 100 Hz; for 20 min per day, once daily, for a total of 7 days. During application, the ultrasonic transducer was placed in water, with the 6-well plates and the acoustic emission surface of the transducer being parallel and positioned above the water surface, with the bottom of the wells contacting the water surface at a vertical distance of 5 cm from the transducer’s emission surface, and the centre of the wells aligned with the centre of the transducer (Figure 1A). The control group, i.e., the sham irradiation group, was placed on the water surface under the same conditions and exposed to a deactivated ultrasonic transducer for sham irradiation for 20 min per day, once daily, for a total of 7 days. Both the LIPUS and control groups had their medium changed after the fourth day of loading.

### 2.5. Exosome Extraction

Preparation of supernatant samples: On the fourth day of ultrasound application and within 24 h of the last application, the cell culture media from the LIPUS and control groups were collected and centrifuged at 300× *g* at 4 °C for 10 min, followed by centrifugation at 2000× *g* at 4 °C for 30 min to remove cells and cell debris. The supernatant was then transferred to new centrifuge tubes and stored at −80 °C.

Ultracentrifugation method for exosome extraction: The cell-free supernatant was collected in sufficient quantities from multiple experiments, centrifuged at 10,000× *g* at 4 °C for 30 min, transferred to centrifuge tubes, filtered through a 0.22 µm filter, transferred to pre-sterilised ultracentrifuge tubes, and centrifuged at 100,000× *g* at 4 °C for 70 min. The supernatant was discarded, the pellet was washed with PBS and centrifuged again at 100,000× *g* at 4 °C for 70 min, the supernatant was discarded, and the pellet was resuspended in PBS and stored at −80 °C (Figure 1A).

### 2.6. Exosome Characterisation

Transmission electron microscopy (TEM, HT7700, Hitachi, Tokyo, Japan) was used to observe and record the morphology of the exosomes. The exosome samples were diluted with PBS to an appropriate concentration, and 20 µL of the suspension was dropped onto a copper grid, allowed to adsorb naturally for 5–10 min, and then dried. A 20 µL drop of 2% phosphotungstic acid solution was added to the copper grid, left to stain for 3–5 min, and then dried and photographed.

The exosome samples were analysed using a ZetaView PMX 110 nanoparticle tracking analyser (Particle Metrix, Meerbusch, Germany) and ZetaView 8.04.02 software (Particle Metrix, Meerbusch, Germany), which recorded the particles’ Brownian motion and measured their size and concentration.

Exosomes were lysed with RIPA buffer (BiYunTian Biotechnology, Shanghai, China). Protein concentration was determined using a BCA Protein Assay Kit (BiYunTian Biotechnology, Shanghai, China). Proteins were transferred to polyvinylidene fluoride (PVDF, Millipore, Darmstadt, Germany) membranes and blocked with 5% non-fat milk (Yili Co., Ltd., Inner Mongolia, China) in TBST (25 mM Tris, 140 mM NaCl, 0.1% Tween 20, pH = 7.5) at room temperature for 2 h. The membrane was incubated overnight at 4 °C with primary antibodies against TSG101 (diluted 1:5000) (Abcam, Cambridge, MA, USA), HSP70 (diluted 1:1000) (Abcam, Cambridge, MA, USA), and calnexin (diluted 1:3000) (Abcam, Cambridge, MA, USA). The membranes were washed three times with TBST, each time for 10 min, and then incubated for 1 h at room temperature with HRP-conjugated goat anti-rabbit IgG secondary antibody (Santa Cruz Biotechnology, Dallas, TX, USA). After three washes with TBST, each lasting one hour, the membranes were treated with an ECL detection reagent (Tanon5200, Shanghai Tianneng Technology Co., Ltd., Shanghai, China) and exposed to obtain protein band images (Figure 1A).

### 2.7. Exosome Uptake Experiment

Exosomes were resuspended in DMEM/F12 medium to a concentration of 1 × 10^9^ particles/well and added to a 24-well plate with 2 mL of complete Dil cell membrane red fluorescent staining solution (BiYunTian Biotechnology Co., Ltd., Shanghai, China) per well. Dil dye was co-cultured with exosomes for 12 h. Then, Dil-labelled exosomes were cultured with HUVEC cells for 24 h, followed by fixation with 4% paraformaldehyde for 10 minutes. Cells were washed three times with PBS for 3 min each, and then 400 μL/well of phalloidin working solution was added to stain the cytoskeleton (Boster Biological Engineering Co., Ltd., Wuhan, China) for 40–60 min. After three washes with PBS, the cell nuclei were stained with 4′,6-diamidino-2-phenylindole (DAPI, Boster Biological Engineering Co., Ltd., Wuhan, China) for 5 min and washed three more times with PBS. All procedures were performed in the dark, and the samples were observed and photographed under a laser confocal microscope (A1, Nikon, Tokyo, Japan) (Figure 2).

### 2.8. Cell Proliferation Experiment

HUVEC cells were seeded at a density of 2000 cells/well in a 96-well plate. After cell adherence, the original medium was replaced with a complete medium containing 1 × 10^9^ exosomes/mL. At 12, 24, 48, and 72 h of co-culture, 10 μL of CCK8 reagent (APExBIO, Houston, TX, USA) was added to each well and incubated for 1 h at 37 °C in a 5% CO_2_ incubator. The optical density at 450 nm was measured using an enzyme-linked immunosorbent assay reader (InfiniteF200, Tecan Trading AG, Männedorf, Switzerland), the values were recorded, and a cell proliferation curve was plotted.

### 2.9. Cell Migration

A total of 5 × 10^4^ HUVEC cells were seeded in the upper chamber of a Transwell 24-well plate (8 μm, Corning, NY, USA), and 600 μL of complete medium containing 1 × 10^9^ exosomes/mL was added to the lower chamber. The chamber was incubated at 37 °C in a 5% CO_2_ incubator for 24 h, after which the inner layer cells of the chamber were gently removed with a moistened cotton swab. The chamber was then fixed in 4% paraformaldehyde for 20 min. After three washes with PBS, the chamber was stained with 1% crystal violet solution (BiYunTian Biotechnology Co., Ltd., Shanghai, China) for 30 min. The chamber was washed three more times with PBS to remove excess dye, and cells were observed and photographed under an inverted microscope.

### 2.10. Endothelial Tube Formation Assay

Matrigel basement membrane matrix (Corning, NY, USA) was coated at a concentration of 50 μL/well on a 96-well plate. 2 × 10^4^ HUVEC cells/well were added, and an exosome solution was introduced to the experimental group at a concentration of 1 × 10^9^/mL. A 100 μL cell suspension was seeded onto the solidified Matrigel gel and incubated at 37 °C in a 5% CO_2_ incubator. The observation began at 3 h post-seeding and continued every 1 h until tube-like structures formed, which were then quantified.

### 2.11. Statistical Analysis

All data are presented as mean ± standard error. Image analysis for the Transwell cell migration assay and tube formation assay was performed using Image J 1.54d software, and data processing was carried out with SPSS version 27.0.1. Comparisons between the two groups were made using an independent samples *t*-test, and for non-normally distributed data, the Mann–Whitney test was used. Comparisons among multiple groups were performed using one-way analysis of variance (one-way ANOVA), with post hoc comparisons using the LSD test. Graphical representations of statistical results were created using GraphPad Prism 8 software. A *p*-value < 0.05 was considered statistically significant, while a *p*-value > 0.05 was not considered statistically significant.

## 3. Results

### 3.1. Isolation and Identification of Exosomes

MC3T3-E1 cells were seeded onto scaffold materials in a 6-well plate. After 7 days of ultrasonic loading, 6 mL of the cell supernatant could be collected from each well. From every 400 mL of cell supernatant, approximately 400 microliters of exosomes could be extracted. Examination under a transmission electron microscope revealed that in both the LIPUS group and the control group, double-layered lipid bilayer vesicles were observed. These exhibited a characteristic “cup-shaped” morphology with a concave middle and a raised edge, ranging in size from 30–150 nm, displaying the typical features of exosomes (Figure 1B). TSG101 and HSP70 are marker proteins indicative of the exosome structure, while calreticulin, a protein that binds calcium ions, is not expressed in exosomes. The expression of TSG101 and HSP70 proteins was detected in both groups of exosomes via the Western blot technique, with no expression of calreticulin observed (Figure 1D).

Utilising nanoparticle tracking analysis (NTA), the particle size distribution and concentration of three batches of samples extracted at different times were analysed. The results indicated that the particle sizes of the exosomes from both the LIPUS group and the control group were distributed within the range of 30–300 nm, with the majority of the particles concentrated between 100 and 200 nm, exhibiting an unimodal distribution with the peak at around 150 nm. This is in line with the typical characteristics of exosomal size distribution. In addition, the range of particle sizes of exosomes from the three different batches did not vary significantly, showing a consistent overall distribution trend (Figure 1C). Based on the NTA results, statistical analysis was conducted on the original concentration of particles in the three batches of samples. The findings demonstrated that the original concentration of exosome samples from both the LIPUS and control groups was around 8 × 10^9^ particles/mL, with no statistically significant difference in concentration between the two samples. The variation in exosome concentrations among the three batches was not pronounced (Figure 1E).

### 3.2. Exosomes Labelled Were Taken up by HUVEC Cells

Observation under confocal laser scanning microscopy revealed a large amount of red Dil fluorescence signal inside HUVEC cells co-cultured with exosomes pre-stained with Dil red cell membrane fluorescence dye, whereas the control group without exosomes showed no red fluorescence signal, indicating that exosomes from both the LIPUS group and the control group were taken up by HUVEC cells (Figure 2).

### 3.3. MEC3T3-Exo Promotes Endothelial Cell Proliferation

To investigate the impact of exosomes secreted by MC3T3-E1 cells under ultrasound on the proliferation ability of HUVEC cells, two types of exosomes at 109 particles/mL were co-cultured with HUVEC cells for 12, 24, 48, and 72 h. The results showed that after 12 h of co-culture, both types of exosomes had no significant effect on HUVEC cell proliferation (*p* > 0.05). After 24 h of co-culture, exosomes from both LIPUS and control groups promoted HUVEC cell proliferation (*p* < 0.05). After 48 h of co-culture, exosomes from the LIPUS group significantly increased HUVEC cell proliferation (*p* < 0.05), while the effect of the control group was not significant. After 72 h of co-culture, the control group also exhibited the ability to promote cell proliferation similar to the LIPUS group (*p* < 0.05). Although there was no significant statistical difference between the two exosome groups, the LIPUS group consistently demonstrated a stronger proliferation-promoting potential compared to the control group, especially at 48 h of co-culture (*p* > 0.05). The results demonstrate that exosomes from both the LIPUS and control groups can promote HUVEC cell proliferation (Figure 3B).

### 3.4. MC3T3-Exo Stimulated by Ultrasound Better Promotes Endothelial Cell Migration

Transwell chambers can be used to evaluate the migration ability of HUVEC cells, with HUVEC cells seeded in the upper chamber and solutions of the two exosome groups at a concentration of 10^9^ particles/mL placed in the lower chamber. As shown in the figure (Figure 3C), 24 h later, both LIPUS group exosomes and control group exosomes could stimulate HUVEC cells to migrate through the small holes at the bottom of the upper chamber (*p* < 0.05). Compared to the control group, LIPUS group exosomes exhibited a stronger ability to promote HUVEC cell migration, with a statistically significant difference (*p* < 0.05) (Figure 3D).

### 3.5. MC3T3-Exo Stimulated by Ultrasound Better Promotes Tube Formation

Tube formation ability indicates that endothelial cells in good condition and appropriate numbers can form vessel-like structures on the surface of a matrix under suitable conditions, which is an important indicator for evaluating endothelial cell function. Under the microscope, after co-culturing for 3 h, a large amount of cell stacking was observed in the control group without exosomes, with no tube-like structures formed; some branching structures were visible in the control group, but no complete tubular structures were observed, while the LIPUS group showed numerous branching structures, with some connecting to form tubes. After co-culturing for 5 h, branching structures appeared in the control group, with a few connecting to form tubes, but no complete tube-like structures were observed. In contrast, the LIPUS group exhibited a dense distribution of numerous small tube-like structures. After co-culturing for 7 h, all three groups showed signs of apoptotic tube formation, with enlarged lumens and reduced numbers of wall cells, but the LIPUS group and control group still maintained relatively intact tube-like structures (Figure 4A).

Image J 1.54d software was used to analyse the microscope images taken after 5 h of co-culture, and statistical analysis was performed on the total tube length, branch length, number of nodes, number of junctions, and number of branches. The results showed that compared to the group without exosomes (control group), the LIPUS group and mock-irradiated HUVEC cells had longer total tube length and branch length (*p* < 0.001), as well as higher node numbers, junction numbers, and branch numbers (*p* < 0.001). When comparing the LIPUS group with the control group, LIPUS group exosomes increased the total tube length (*p* < 0.05), number of nodes (*p* < 0.05), number of junctions (*p* < 0.01), and number of branches (*p* < 0.05) in HUVEC cell tube formation. However, there was no statistical difference between the two groups in branch length or tube formation (*p* > 0.05) (Figure 4B).

## 4. Discussion

Our previous results proved that LIPUS facilitated cellular ingrowth and enhanced the proliferation and early differentiation of osteoblasts and bone formation in Ti-6Al-4V scaffolds [18,19,20]. In the process of bone formation, angiogenesis plays a pivotal role because blood vessels not only act as a source of oxygen and nutrients but also facilitate the transportation of bone progenitor cells, supply essential minerals like calcium and phosphate for mineralisation, and contribute to the establishment of a niche for bone marrow stromal cells and hematopoietic stem cells. Therefore, studying how osteoblasts on Ti-6Al-4V scaffolds under LIPUS stimulation affect angiogenesis is essential to confirm their potential for repairing bone defects.

Contrary to the traditional belief that angiogenesis solely influences bone remodelling, recent research has underscored the intricate spatial and temporal correlation between bone formation and angiogenesis, termed “angiogenesis–bone coupling”. This coupled process has been proven to be related to paracrine mechanisms, with cytokines playing a significant role. For example, osteoblast-derived VEGF is essential for the early vascularisation response and macrophage infiltration in the initial stages of inflammation and later stimulates angiogenesis and osteoblast differentiation [21]. MicroRNAs (miRNAs) have also been shown to control angiogenesis at multiple levels, expressed within vascular constituents, and play key roles in development and disease [22]. This paracrine-mediated interplay offers insight into the potential function of exosomes in this process. Exosomes derived from bone marrow mesenchymal stem cells have been demonstrated to promote angiogenesis–osteogenesis coupling both in vitro and in vivo [14].

The interaction between LIPUS and exosomes has become an intriguing area of study, with several studies suggesting that LIPUS can modulate the release, content, and uptake of exosomes, thereby affecting cellular communication and tissue regeneration. Both in vitro and in vivo experimental results demonstrate that extracellular vesicles secreted by LIPUS-treated BMSCs possess stronger anti-inflammatory properties compared to extracellular vesicles secreted by unstimulated BMSCs [23]. However, to date, there have been no reports on the role of LIPUS in influencing communication between osteoblasts and endothelial cells through the effect on exosomes secreted by MC3T3-E1 cells on porous Ti-6Al-4V scaffolds.

In our experiments, following co-culture with Dil red fluorescent cell membrane dye, internalisation of exosomes by HUVECs was indicated by the presence of red fluorescent signals within the cells. The intensity of these signals increased in correlation with the concentration of exosomes and the duration of co-culture, suggesting that HUVECs are capable of taking up exosomes from the solution in a sustained manner, demonstrating a time-dependent and concentration-dependent uptake effect.

After internalisation by cells, exosomes can exert certain biological functions. This study found that the exosomes from both the LIPUS and control groups promoted proliferation, migration, and tube formation of HUVECs, which may be related to the paracrine effects of MC3T3-E1 cells on endothelial cell biological functions. Studies have confirmed that MC3T3-E1 cells can promote the recruitment and proliferation of endothelial cells at fracture sites by secreting vascular endothelial growth factor (VEGF) and angiogenic factors [24]. Co-culture of HUVECs with conditioned medium from osteoblasts revealed that it could enhance endothelial cell proliferation, migration, and differentiation capabilities [25]. Exosomes derived from senescent osteoblasts mediated by miR-139-5p can promote aging and apoptosis by targeting the TBX1 gene, thereby inhibiting the proliferation and invasion of endothelial cells [11]. Building upon this foundation, our experiments confirmed that exosomes from MC3T3-E1 cells could play a positive role in the coupling process of angiogenesis and osteogenesis.

Endothelial cell migration, an early step in the angiogenesis cascade and a hallmark of angiogenesis, occurs autonomously but can acquire collective migration characteristics upon stimulation, when a group of cells migrate towards the chemotactic stimulus and the remaining cells follow in the same direction. This process is crucial for stimulating angiogenesis in tissue regeneration [26]. Transwell assays in various experiments involving the co-culture of exosomes and HUVECs were employed to assess the migratory impact of exosomes from the lower chamber on HUVECs in the upper chamber [27,28,29]. Our study utilised two sets of exosomes in the lower chamber of the transwell system to investigate their chemotactic effect on HUVECs in the upper chamber, and the results demonstrated that the LIPUS group was more effective in driving the migration of HUVECs towards the lower chamber compared to the control group (*p* < 0.01), thereby enhancing the migratory capacity of HUVECs.

The basement membrane is a thin, highly specialised extracellular matrix that serves as a foundation for endothelial cells and maintains vascular morphology in vivo. In vitro, when endothelial cells are seeded on recombinant basement membrane matrices, they rapidly organise into tubular structures. This principle has been adapted into a matrix gel, which is a convenient and quantifiable method for assessing the tube-forming ability of endothelial cells [30]. Our experiments found that the LIPUS group formed microtubes with complete luminal structures more rapidly compared to the control group and exosome-free control group, with a greater density of microtubes. Quantitative analysis using Image J 1.54d software revealed that the total length of microtubes, number of branching microtubes, number of nodes, and number of connecting points were all significantly higher in the LIPUS group than in the control group, which proves that exosomes from MC3T3-E1 cells under ultrasound can promote endothelial cell tube formation. These experimental outcomes corroborate the role of ultrasound-modulated exosomes secreted by MC3T3-E1 cells in enhancing the biological functions of endothelial cells.

## 5. Conclusions

In this study, MC3T3-E1 cells produced exosomes on porous Ti-6Al-4V scaffold materials under LIPUS loading. Compared to the non-ultrasound group, the LIPUS group enhanced endothelial cell proliferation and angiogenesis through exosomes secreted by MC3T3-E1 cells. The LIPUS group formed microtubes with complete luminal structures more rapidly, exhibiting greater density, increased total length, and a higher number of branching microtubes, nodes, and connecting points. Therefore, LIPUS may be a promising strategy to boost the efficacy of osteoblast-derived exosomes in porous Ti-6Al-4V scaffold materials to enhance angiogenesis. However, the effectiveness of osteoblast-derived exosomes in promoting angiogenesis within these scaffold materials remains unclear and requires further investigation.

## Figures and Tables

**Figure 1 biomolecules-14-01154-f001:**
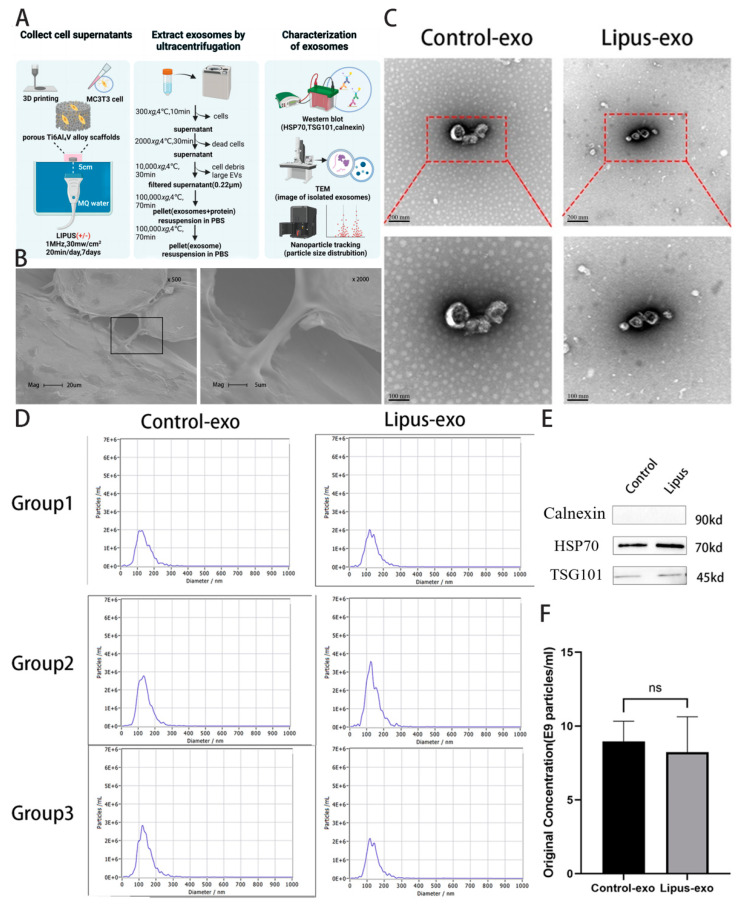
Isolation and identification of exosomes. The separated exosomes were characterised by scanning electron microscopy, nanoparticle tracking analysis (NTA), and identification of surface marker proteins. (**A**) Methodology for ultrasonic loading of cells and extraction and identification of exosomes. (**B**) SEM images of MC3T3-E1 were grown within the porous Ti-6Al-4V scaffold. (**C**) Morphology of exosomes under electron microscopy, scale bar = 200 nm. (**D**) Particle size distribution and concentration in the exosome suspension, with the size of particles in both LIPUS and control groups ranging between 30 and 300 nm, consistent with known exosomal dimensions. (**E**) Expression of HSP70, TSG101, and calreticulin in exosomes from each group. Western blot original images can be found in Appendix A. (**F**) Measurement of exosome concentration in three independent experiments showed no significant difference between the LIPUS and control groups. ns: *p* ˃ 0.05.

**Figure 2 biomolecules-14-01154-f002:**
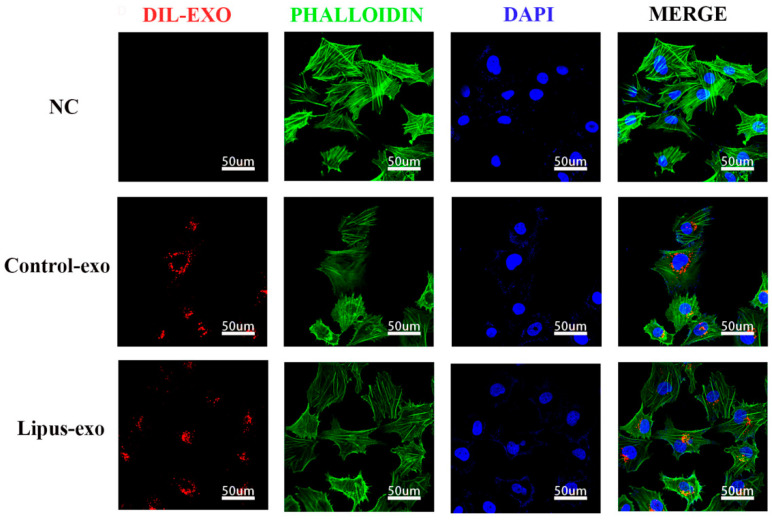
Endothelial cells took up Dil-labelled exosomes. After co-culturing with pre-stained exosomes for 24 h, red fluorescence signals could be observed inside both LIPUS and control group HUVEC cells, with no significant difference in fluorescence signal intensity between the two groups.

**Figure 3 biomolecules-14-01154-f003:**
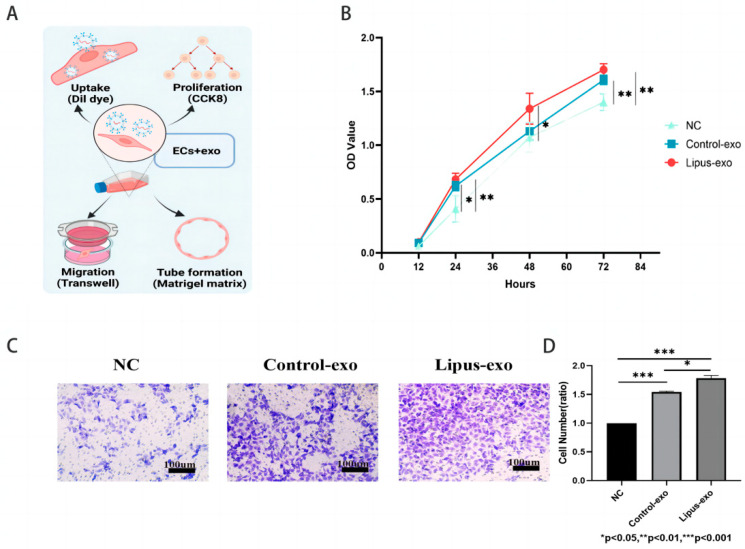
Exosomes extracted from MC3T3-E1 cells stimulated by ultrasound better promote endothelial cell migration. (**A**) Schematic of functional experiments. (**B**) CCK8 assay to assess endothelial cell proliferation. (**C**) Transwell assay to assess exosome-promoted endothelial cell migration ability, scale bar = 100 μm. (**D**) Statistical analysis results of endothelial cell migration. One-way analysis of variance was used for testing, and pairwise comparisons were conducted using a post hoc LSD test, n = 3, *: *p* < 0.05, **: *p* < 0.01, ***: *p* < 0.001.

**Figure 4 biomolecules-14-01154-f004:**
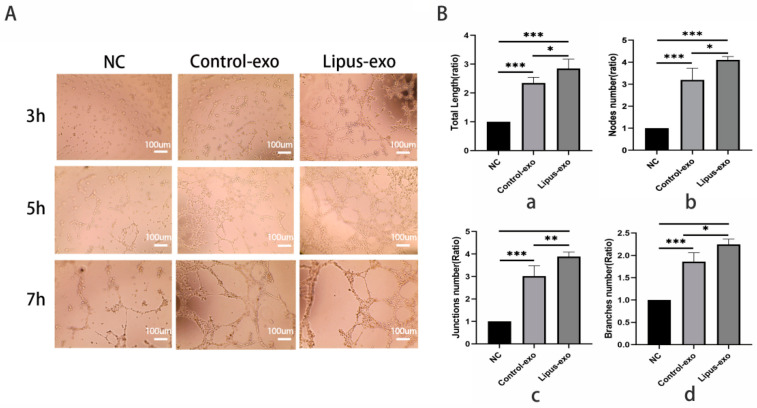
Endothelial cell tube formation experiment. (**A**) Under microscopic observation, scale bar = 100 um. (**B**) Image J was used for statistical analysis of the tube formation experiment results: (a) total tube length, (b) number of nodes, (c) number of junctions, (d) number of branches. One-way analysis of variance was used for testing, and pairwise comparisons were conducted using a post hoc LSD test, n = 3, *: *p* < 0.05, **: *p* < 0.01, ***: *p* < 0.001.

## Data Availability

Data are contained within the article.

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
