# Peer review of "3D-Cultured MC3T3-E1-Derived Exosomes Promote Endothelial Cell Biological Function under the Effect of LIPUS"

_biomolecules, 2024, doi:10.3390/biom14091154_

Round 1
Reviewer 1 Report
Comments and Suggestions for Authors
In this study, exosomes produced by osteoblasts on porous Ti-6Al-4V scaffold materials under LIPUS and non-ultrasound stimulated conditions were co-cultured with endothelial cells. Compared to the non-ultrasound group, the LIPUS group facilitated endothelial cell proliferation and angiogenesis.
At the end of the Introduction section, authors should clearly define the objective of the study. In the present form, it is very confusing to the reader.
The manuscript does not present enough information concerning the porous Ti-6Al-4V scaffold used to culture the cells – specifically, SEM images proving the porosity of the scaffold will be useful. Authors referred a previous work, however, the reference was not provided.
The porous scaffold was seeded with Mouse pre-osteoblast precursor cells MC3T3-E1 – SEM images are mandatory to prove that the cells were growing within the porous structure.
In the Abstract, authors claimed that “The findings indicated that the exosomes were consistently and stably uptook by the endothelial cells on porous Ti-6Al-4V scaffold materials”. However, in the Methodology section, it appears that Human umbilical vein endothelial cells (HUVEC) were cultured in standard culture plates in the presence of exosomes (not on the porous scaffold). Regarding this, the description in section 2.7. Exosome Uptake Experiment is confusing, namely the sentence: “The exosomes were co-cultured with the cells for 12 hours, labeling the MC3T3-exo. HU-173 VEC cells were cultured with Dil-labeled exosomes for 24 hours, followed by fixation with 174 4% paraformaldehyde for ten minutes”. Please, clarify.
In the Discussion section, the first sentence needs the respective reference. The second and third paragraphs should be greatly summarized, since the referred information does not contribute specifically to the discussion of the results reported in this study.
The information provided by this manuscript has a major limitation: exosomes were collected from a mouse cell line and were used to treat human endothelial cells. Therefore, the conclusions may be doubtful.
Reviewer 2 Report
Comments and Suggestions for Authors
Dear authors,
Thank you for submitting your manuscript titled ‘Exosomes derived from 3D cultured MC3T3-E1 promote the biological function of the endothelial cell under the effect of LI-3 PUS’ to ‘Biomolecules’. After careful consideration and thorough review, the manuscript can be accepted for publication after minor revision.
1- The title could be change to ‘3D-cultured MC3T3-E1-derived exosomes promote endothelial cell biological function under the effect of LIPUS.’
2- In the introduction section, the aim of the work is not clear. The authors should clarify the role of the scaffold in promoting angiogenesis and its crucial rule in the process of bone formation
3- In paragraph 2.1 the author wrote: “A cylindrical three-dimensional porous Ti-6Al-4V scaffold with a diameter of 15mm, 85 height of 2mm, pore size of 400-500 micrometres, and porosity of 65%-70% was manufac-86 tured using an electron beam melting system (Arcam A1, Arcam, Sweden) as described in 87 our previous work”. Reference of their previous work should be added.
4- Authors should correct 1x109 exosomes/ml in 1x109 exosomes/ml in all parts where they write about exosome concentration as well as for cells (e.g. in line 203).
5-.There are no results regarding the cylindrical three-dimensional porous Ti-6Al-4V scaffold as a material characterization and a representative image. The authors should be added these information.
6- If the figure 1A was sourced from an online publication, please provide a citation. In addition, the authors should increase the resolution of both figure 1A and C. In particular, figure 1C is completely illegible.
7- The scale bar dimension in figure 2 was reported only in Lipus exo (Merge), the authors should add it in the other images too. The same think for the figure 3C where the scale bar dimension was reported only for Lipus exo.
8- The legend of figures 3 and 4 seems to be the description of the results. The authors should shorten it by limiting themselves to the description of the figure.
9- The authors should stretch the conclusions. They could use some sentences from the last part of the discussion so as not to be repetitive.
Round 2
Reviewer 1 Report
Comments and Suggestions for Authors
The authors have addressed my comments, making the manuscript clearer for readers.